

# *Escherichia coli* alcohol dehydrogenase YahK is a protein that binds both iron and zinc

Feng Liang[1],[*], Shujuan Sun[2],[*], YongGuang Zhou[3], Tiantian Peng[3], Xianxian Xu[3], Beibei Li[3] and Guoqiang Tan[3]

[1] Department of Clinical Laboratory, The First Affiliated Hospital of Wenzhou Medical University, Key Laboratory of Clinical Laboratory Diagnosis and Translational Research of Zhejiang Province, Wenzhou, Zhejiang, China
[2] Shandong Provincial Key Laboratory of Detection Technology for Tumor Markers, College of Medicine, Linyi University, Linyi, Shandong, China
[3] Laboratory of Molecular Medicine, Zhejiang Provincial Key Laboratory for Technology and Application of Model Organisms, Key Laboratory of Laboratory Medicine, Ministry of Education, China, School of Laboratory Medicine and Life Sciences, Wenzhou Medical University, Wenzhou, Zhejiang, China
[*] These authors contributed equally to this work.

Corresponding author
Guoqiang Tan, tgq@wmu.edu.cn

## ABSTRACT

**Background:** Previous studies have highlighted the catalytic activity of *Escherichia coli* alcohol dehydrogenase YahK in the presence of coenzyme nicotinamide adenine dinucleotide (NAD) and metal zinc. Notably, competitive interaction between iron and zinc ligands has been shown to influence the catalytic efficiency of several key proteases. This study aims to unravel the intricate mechanisms underlying YahK's catalytic action, with a particular focus on the pivotal roles played by metal ions zinc and iron.

**Methods:** The purified YahK protein from *E. coli* cells cultivated in LB medium was utilized to investigate its metal-binding properties through UV-visible absorption measurements and determination of metal content. Subsequently, the effects of excess zinc and iron on the metal-binding ability and alcohol dehydrogenase activity of the YahK protein were explored using M9 minimal medium. Furthermore, site-directed mutagenesis technology was employed to determine the iron-binding site location within the YahK protein. Polyacrylamide gel electrophoresis was conducted to examine the relationship between iron and zinc with respect to the YahK protein.

**Results:** The study confirmed the presence of iron and zinc in the YahK protein, with the zinc-bound form exhibiting enhanced catalytic activity in alcohol dehydrogenation reactions. Conversely, the presence of iron appears to play a pivotal role in maintaining overall stability of the YahK protein. Furthermore, experimental findings indicate that excessive zinc within M9 minimal medium can competitively bind to iron-binding sites on YahK, thereby augmenting its alcohol dehydrogenase activity.

**Conclusion:** The dynamic binding of YahK to iron and zinc unveils its intricate regulatory mechanism as an alcohol dehydrogenase, thereby highlighting the possible physiological role of YahK in *E. coli* and its significance in governing cellular metabolic processes. This discovery provides a novel perspective for further

investigating the specific impact of metal ion binding on YahK and *E. coli* cell metabolism.

# INTRODUCTION

Alcohol dehydrogenase (ADH) is an important redox catalyst in living organisms, present in various organisms from bacteria to humans. It catalyzes the reversible oxidation of various primary and secondary alcohols to the corresponding aldehydes and ketones with the participation of coenzyme $NAD^+$ ($NADP^+$) (*An, Nie & Xu, 2019*; *Chen, 1995*; *Forrest & Gonzalez, 2000*; *Hall & Bommarius, 2011*; *Nguyen et al., 2020*). The ADH superfamily can be categorized into three groups: zinc-dependent medium-chain ADHs (about 350 residues per subunit); short-chain zinc-independent ADHs (about 250 residues per subunit); and "iron-activated" ADH (each subunit is approximately 385 residues in size) (*Cho et al., 2022*; *Reid & Fewson, 1994*). *Escherichia coli* alcohol dehydrogenase YahK consists of 349 amino acids, is a typical medium-chain dehydrogenase, and belongs to the cinnamyl alcohol dehydrogenase family (CADs) (*Nordling, Jörnvall & Persson, 2002*; *Pick et al., 2013*). Previous studies speculated that YahK was a zinc-containing oxidoreductase involved in the transformation of *E. coli* from an aerobic to facultative anaerobic state (*Lopez-Campistrous et al., 2005*). Numerous investigations have demonstrated the pivotal role of YahK in processes such as biofilm formation, maturation, regulation of NADH equilibrium, and metabolic engineering (*Koma et al., 2012*; *May & Okabe, 2011*; *Zhang et al., 2011*). Upon scrutinizing the sequence's structural attributes, the researchers found that *E. coli* YahK features a zinc-binding motif with the arrangement $GHEX_2GX_5$ (G/A)$X_2$ (I/V/A/C/S) and a coenzyme binding domain characterized by a $GX_{1-3}GX_{1-3}G$ sequence. These features are similar to zinc-containing medium-chain ADH (*Persson, Hedlund & Jörnvall, 2008*; *Sulzenbacher et al., 2004*). Specifically, Cys40, His62, and Cys158 within the YahK protein serve as active zinc binding sites, exhibiting catalytic properties. In contrast, Cys93, Cys96, Cys99, and Cys107 function as structural zinc binding sites, thereby influencing subunit interactions and sustaining structural stability. It is worth mentioning that a crystallographic representation of YahK can be accessed in the Protein Data Bank (PDB) under the entry 1UUF, although additional information about this structure is not readily available (*Cho et al., 2022*; *Jeudy, Claverie & Abergel, 2004*; *Jörnvall, Persson & Jeffery, 1987*; *Nordling, Jörnvall & Persson, 2002*; *Pick et al., 2013*).

The presence of macroelements and microelements is indispensable for the normal physiological functioning of organisms, as they play pivotal roles in various biological processes encompassing cellular metabolism, antioxidant and anti-inflammatory defenses. Moreover, these elements exert regulatory effects on enzyme activity, gene expression modulation, and active participation in protein synthesis (*Choi et al., 2016*; *Grzeszczak, Kwiatkowski & Kosik-Bogacka, 2020*). Zinc is an essential trace metal in living organisms (*Roohani et al., 2013*). Zinc ensures the correct folding of proteins and plays a crucial role

in the enzymatic catalysis process (*Berg & Shi, 1996*). In most living organisms, zinc is the most abundant metal element after iron (*Berg & Shi, 1996*). In *E. coli* grown in LB medium, zinc could accumulate to levels similar to iron, calcium, *etc.*, (~0.2 mM) (*Outten & O'Halloran, 2001*; *Sevcenco et al., 2011*). Removing zinc from the medium produced a slow-growing phenotype and activated its zinc uptake system (*Graham et al., 2009*). Iron and zinc have been reported to have similar ligand-binding coordination in proteins (*Dauter et al., 1996*). Studies have revealed that the enzymatic activity of *E. coli* YahK was influenced by the presence of zinc at specific binding sites (*Pick et al., 2013*). Despite knowledge about the significant impact of active zinc and structural zinc binding sites on both catalytic function and structural stability of YahK, the precise mechanisms by which these metal ions regulate its enzyme activity and adapt *E. coli* to different environmental conditions remain unclear. Therefore, further research is imperative to elucidate the detailed regulatory mechanisms employed by these metal ions in modulating YahK's activity and their profound effects on metabolic regulation and environmental adaptability of *E. coli*. This holds great significance for comprehending bacterial metabolic pathway regulation, optimizing metabolic engineering strategies, and developing novel antimicrobial treatments. Our investigation uncovered that *E. coli* YahK is capable of binding both zinc and iron. However, the zinc-bound form of YahK shows robust catalytic activity in alcohol dehydrogenation, whereas the iron-bound variant exhibits limited ability in facilitating this process. Furthermore, iron may play a role in upholding the protein's stability.

## MATERIALS AND METHODS

### Protein purification

The alcohol dehydrogenases YahK and AdhP from *E. coli* are zinc-dependent medium-chain ADHs, consisting of 349 and 336 amino acids, respectively (*Pick et al., 2013*; *Thomas et al., 2013*). Both proteins exhibit a zinc-binding GHEX$_2$GX$_5$ (G/A)X$_2$ (I/V/A/C/S) protein motif, as well as the GX$_{1-3}$GX$_{1-3}$G pattern located in the nucleotide-binding region (*Pick et al., 2013*; *Thomas et al., 2013*). Protein alignment analysis revealed a significant similarity of 27% with an e-value of 2e$^{-29}$ between YahK and AdhP. Therefore to facilitate the study of *E. coli* alcohol dehydrogenases YahK metal-binding properties and activity, the well-studied AdhP was used as a reference in this study. The wild-type *E. coli* genomic DNA was utilized as the template, while plasmid pET28b$^+$ (Novagen, Darmstadt, Germany) served as the expression vector. The YahK and AdhP genes were amplified through polymerase chain reaction using primers containing oligonucleotides with *HindIII* and *NcoI* restriction endonuclease site (Table 1). Recombinant YahK and AdhP were subsequently expressed in the *E. coli* BL21 (DE3) strain, cultivated in LB medium.

Purification of the proteins with six histidines at the C-terminus were purified following the previously described protocol (*Tan et al., 2017*). Harvested cells were resuspended in protein preservation solution (buffer A: 500 mM NaCl, 20 mM Tris-HCl, pH 8.0). The histidine-rich recombinant proteins were then purified using a nickel-ion metal-chelated affinity chromatography medium, Ni-NTA. Buffer A served as the running buffer, while buffer B (500 mM NaCl, 20 mM Tris-HCl, 500 mM imidazole, pH 8.0) functioned as the

**Table 1 PCR primers used in this work.**

| Primer | Sequence (5′–3′)[a] | Application |
|---|---|---|
| YahK-1 | TTTAAGAAGGAGATATACCATGAAGATCAAAGCTGTTGGTGC | Vector construction |
| YahK-2 | TGCTCGAGTGCGGCCGCAAGGTCTGTTAGTGTGCGATTATCGATAAC | |
| AdhP-1 | TTTAAGAAGGAGATATACCATGAAGGCTGCAGTTGTTA | Vector construction |
| AdhP-2 | TGCTCGAGTGCGGCCGCAAGGTGACGGAAATCAATCACCAT | |
| YahK-C40A-1 | AAATCGCTTACTGTGGCCGTTGCCCATTCCGATCTC | Site-specific mutation |
| YahK-C40A-2 | GCAACGCCACAGTAAGCGATTTCGATTTTGACAT | |
| YahK-C99A-1 | GTTGTAAACATTGCGAAGAGGCTGAAGACGGGTT | Site-specific mutation |
| YahK-C99A-2 | GCCTCTTCGCAATGTTTACAACTGTCGACAATGC | |

**Note:**
[a] The underlined bases indicate mutation sites.

elution buffer. A second purification step involved size exclusion chromatography (SEC) utilizing a Superdex 200 16/60 gel filtration column from General Electric Healthcare, which was equilibrated with buffer A, excluding imidazole. Protein purity was verified by SDS-PAGE and protein concentration was determined employing a U3900 (Hitachi High-Tech Co., Tokyo, Japan) instrument, utilizing the published theoretical extinction coefficients. The protein concentration was also determined using a bicinchoninic acid protein assay kit (Thermo Fisher Scientific, Waltham, MA, USA), employing bovine serum albumin as the standard. Notably, both methods yielded highly comparable results.

When evaluating the metal binding properties of YahK, M9 minimal medium with clearly defined components was chosen to induce expression (*CSHL, 2010*). Recombinant YahK was expressed in *E. coli* cells grown in M9 minimal medium supplemented with a fixed concentration of ferric citrate (50 μM), glycerol (0.2%), thiamine (5 μg/ml), and 20 amino acids (100 μg/ml) at 37 °C. Different concentrations of zinc sulfate (0 to 8 μM) were added to the M9 minimal medium before protein expression was induced by L-Arab. The absorbance peak amplitude at 330 nm for purified YahK was utilized to assess the extent of iron binding within YahK.

## Alcohol dehydrogenase activity assay

The alcohol dehydrogenase activity assay was conducted according to the protocol described by *Pick et al. (2013)*. Alcohol dehydrogenase utilizes $NADP^+$ as a coenzyme to catalyze the dehydrogenation of alcohol into acetaldehyde. Simultaneously, NADPH is produced, which exhibits a characteristic absorption peak at 340 nm. The enzymatic activity assay reaction mixture consisted of 20 mM ethanol, 2 mM $NADP^+$, 100 mM N,N-Bis(2-hydroxyethyl)glycine (Bicine) at pH 8.5, and 50 mM phosphate buffer at pH 7.4. When measuring ADH enzyme activity, the reaction system for enzyme activity measurement should be thoroughly mixed. Subsequently, 170 μL of the reaction solution was transferred to a cuvette and incubated at 37 °C for 2 min. Then, 30 μL of purified protein was added to the system. It was mixed by inversion, and the monitoring of the change in $OD_{340}$ was initiated immediately. The change in $OD_{340}$ reflects the activity of

alcohol dehydrogenase. The unit of enzyme activity was defined as the quantity of NADPH generated by 1 µM of enzyme during 1 min of enzymatic activity.

## Metal content analysis

The determination of total iron content in protein samples was carried out using the iron indicator FerroZine, following the procedure described earlier (*Cowart, Singleton & Hind, 1993*). Briefly, protein samples were heated to 85 °C for 30 min with 500 µM ferroZine and 2 mM L-cysteine. After incubation, the protein samples were centrifuged for 5 min at 12,000 rpm to remove the precipitate. The concentration of the iron-ferroZine complex was measured at 564 nm using an extinction coefficient of 27.9 $mM^{-1}$ $cm^{-1}$. For the assessment of total zinc content in protein samples, the zinc indicator PAR (4-(2-pyridylazo)-resorcinol) method was employed (*Bae et al., 2004*). To the protein solution under investigation, add 2 mM DTT and 2 mM PAR followed by the addition of hydrogen peroxide with a final concentration of 10 mM. Allow the mixture to incubate at room temperature for 20 min, then centrifuge it at 12,000 rpm for 5 min to remove denatured protein precipitate and collect the supernatant. Measure the absorbance at $OD_{500nm}$ using a UV-visible spectrophotometer and determine the zinc concentration in the solution based on an absorption coefficient of 66.0 $mM^{-1}$ $cm^{-1}$. The quantification of zinc and iron in the purified proteins was accomplished through inductively coupled plasma emission spectrometry (ICP-MS). It is noteworthy that both metal content analyses yielded congruent results.

## Site-directed mutagenesis

The plasmid YahK-pET28b$^{+}$ was previously constructed as described earlier. Based on information obtained from the Ecogene and NCBI databases, it was ascertained that YahK can bind two zinc ions: active zinc binding sites Cys40, His62, and Cys158, as well as structural zinc binding sites Cys93, Cys96, Cys99, and Cys107. Mutant strains involving alterations at the Cys40 and Cys99 sites, where the original amino acids were substituted with alanine, were generated using Mut Express MultiS Fast Mutagenesis Kit (Vazyme Biotech Co., Ltd. Nanjing, China). The PCR primers used in this endeavor are detailed in Table 1, and the specific mutations were corroborated through direct sequencing.

## Statistical analysis

The data analysis was performed using KaleidaGraph (Synergy Software, Inc., Eden Prarie, MN, USA) and GraphPad Prism version 6.0 (GraphPad Software, Inc., La Jolla, CA, USA). Data were expressed as the mean ± standard deviation (mean ± SD) from three independent experiments, and differences between groups were analyzed using an unpaired t-test or one-way ANOVA followed by a Tukey test. $P < 0.05$ was considered statistically significant.

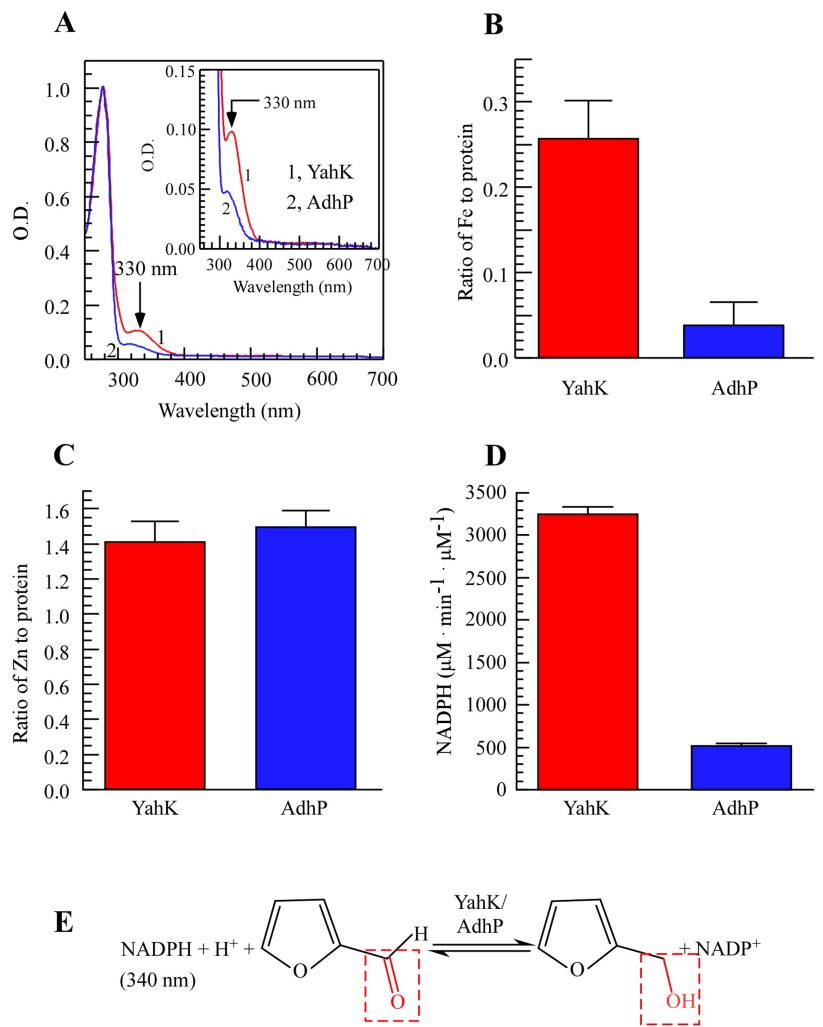

**Figure 1 *E. coli* YahK is an iron and zinc binding aldehyde reductase.** (A) UV-visible absorption spectra of recombinant YahK and a positive aldehyde reductase control, AdhP, purified from *E. coli* cells grown in LB medium under aerobic growth conditions. The concentration of YahK and AdhP were 37.5 μM and 44 μM, respectively. (B and C) Iron and zinc contents of purified YahK and AdhP. (D) The aldehyde reductase activity of purified YahK and AdhP. The samples in (B and C) were the same as in panel (A). The results are the average ± standard deviation from three independent experiments. (E) Schematic illustration of the catalytic transfer hydrogenation process of furfural to furfural alcohol and NADP⁺. The data are presented as the mean ± S.D. $p < 0.05$.

## RESULTS

### *E. coli* alcohol dehydrogenase YahK has an iron-binding affinity

To study the metal-binding properties of YahK, we expressed YahK and AdhP in BL21 (DE3) cells cultured in LB medium under aerobic growth conditions due to their shared zinc-binding domain (*Pick et al., 2013*; *Thomas et al., 2013*). By conducting UV-Vis absorption measurements, a significant absorption peak at 330 nm was observed for YahK compared to AdhP (Fig. 1A), indicating its potential iron-binding capability. Regarding metal content analysis, it was found that each monomeric unit of YahK contained an

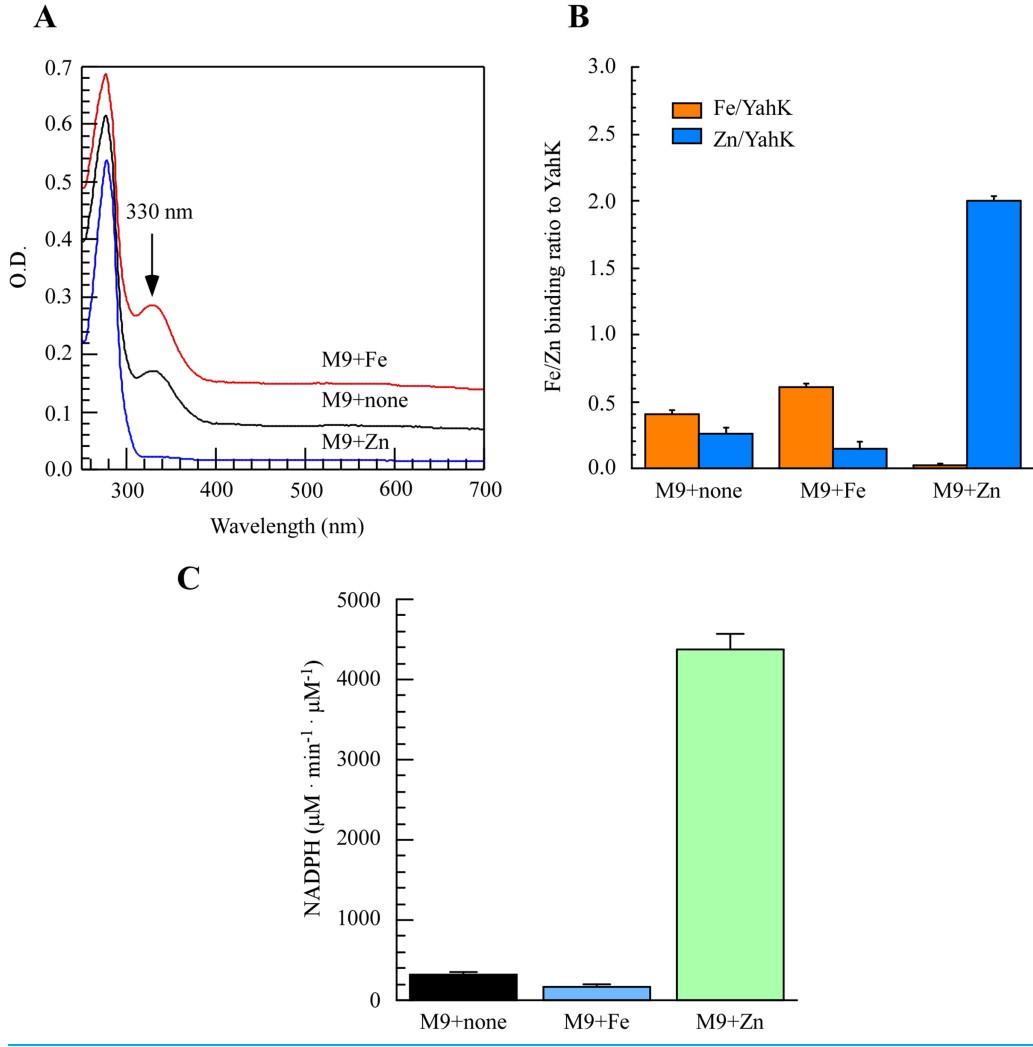

**Figure 2 The iron or zinc binding activity of YahK purified from M9 minimal medium supplemented with exogenous iron or Zinc.** (A) UV-visible absorption spectra of YahK purified from *E. coli* cells grown in M9 minimal medium with no supplementation, 50 μM ferric citrate, or 50 μM zinc sulfate. The protein concentration of purified YahK was 20 μM. (B) Iron and zinc contents of purified YahK from *E. coli* cells grown in M9 minimal medium with no supplementation, 50 μM ferric citrate, or 50 μM zinc sulfate. (C) The relative enzyme activity of purified YahK from *E. coli* cells grown in M9 minimal medium with no supplementation, 50 μM ferric citrate, or 50 μM zinc sulfate. The results are the average ± standard deviation from at least five independent experiments. The data are presented as the mean ± S.D. $p < 0.05$.      

average of $0.26 \pm 0.01$ iron atoms and $1.4 \pm 0.06$ zinc atoms (Figs. 1B and 1C). Furthermore, the iron binding ratio of YahK was significantly higher than that of AdhP, while the zinc binding ratio remained relatively constant. These preliminary observations suggested that *E. coli* YahK was capable of binding to both iron and zinc. Additionally, when compared with the control group's AdhP protein, YahK exhibited significantly higher ADH activity (Fig. 1D).

In conclusion, YahK demonstrated a strong affinity to bind both iron and zinc *in vivo*, with its elevated iron binding ratio showing a positive correlation with increased catalytic activity. Overall, substantial alterations in iron binding may play a pivotal role in enhancing YahK's catalytic activity. Figure 1E depicts the enzymatic reaction mechanism.

## Iron and zinc binding activities of *E. coli* YahK in M9 minimal medium

To directly assess the metal-binding characteristics of proteins and exercise better control over metal content in the growth medium, we opted for M9 minimal medium to induce the expression of the target protein. As depicted in Fig. 2A, purification of YahK from *E. coli* cells cultured in M9 minimal medium under aerobic conditions revealed a prominent absorption peak at 330 nm. Furthermore, analysis of total metal content demonstrated that each YahK monomer contained an average of 0.40 ± 0.02 iron atoms and 0.25 ± 0.04 zinc atoms (Fig. 2B). In parallel experiments, supplementation with additional iron ions (50 μM) to the M9 minimal medium significantly intensified the absorbance peak of YahK at 330 nm, resulting in each YahK monomer contained 0.60 ± 0.02 iron atoms and 0.15 ± 0.04 zinc atoms (Figs. 2A and B). Conversely, the introduction of exogenous zinc (50 μM) into the M9 minimal medium resulted in YahK monomers containing only trace amounts of iron (0.02 ± 0.01 atoms), while being enriched with zinc (2.0 ± 0.04 atoms) (Figs. 2A and B).

We further investigated the effect of different culture conditions on the activity of YahK by using $NADP^+$ as its specific substrate. Experimental findings demonstrate that in the M9 minimal medium supplemented with or without 50 μM ferric citrate, the rate of NADPH generation was relatively low. Figure 2C illustrated a significant increase in the rate of NADPH generation upon addition of zinc sulfate (50 μM), indicating that zinc presence substantially enhances YahK's catalytic efficiency. In summary, our results reveal pronounced iron and zinc binding activity exhibited by YahK in the M9 minimal medium. Specifically, we observed heightened affinity between YahK and iron in its presence, while also noting increased zinc binding affinity alongside notable NADPH generation activity when zinc is present. This result further highlights the apparent correlation between the metal binding capacity of YahK and the specific metal content in the medium.

## Regulation of YahK activity by iron and zinc binding in *E. coli*

To investigate the regulatory role of metal iron and zinc in the activity of YahK within cells, we induced the expression of YahK in the M9 minimal medium containing a fixed iron concentration (50 μM) and gradually increased zinc concentration (0–8.0 μM) under aerobic conditions. In the experiment, as the exogenous zinc concentration in the M9 minimal medium increased, a significant change was observed in the absorption peak at 330 nm. Specifically, as the zinc sulfate concentration increased from 0 to 2.4 μM, there was a sharp rise in the number of zinc atoms per protein monomer while the number of iron atoms gradually decreased and approached zero with increasing zinc concentration. These findings suggested a preference for zinc binding over iron by YahK when both metals were present. Concomitantly, upon raising zinc sulfate concentration from 0 to 2.4 μM, YahK activity exhibited rapid enhancement reaching an apparent plateau,

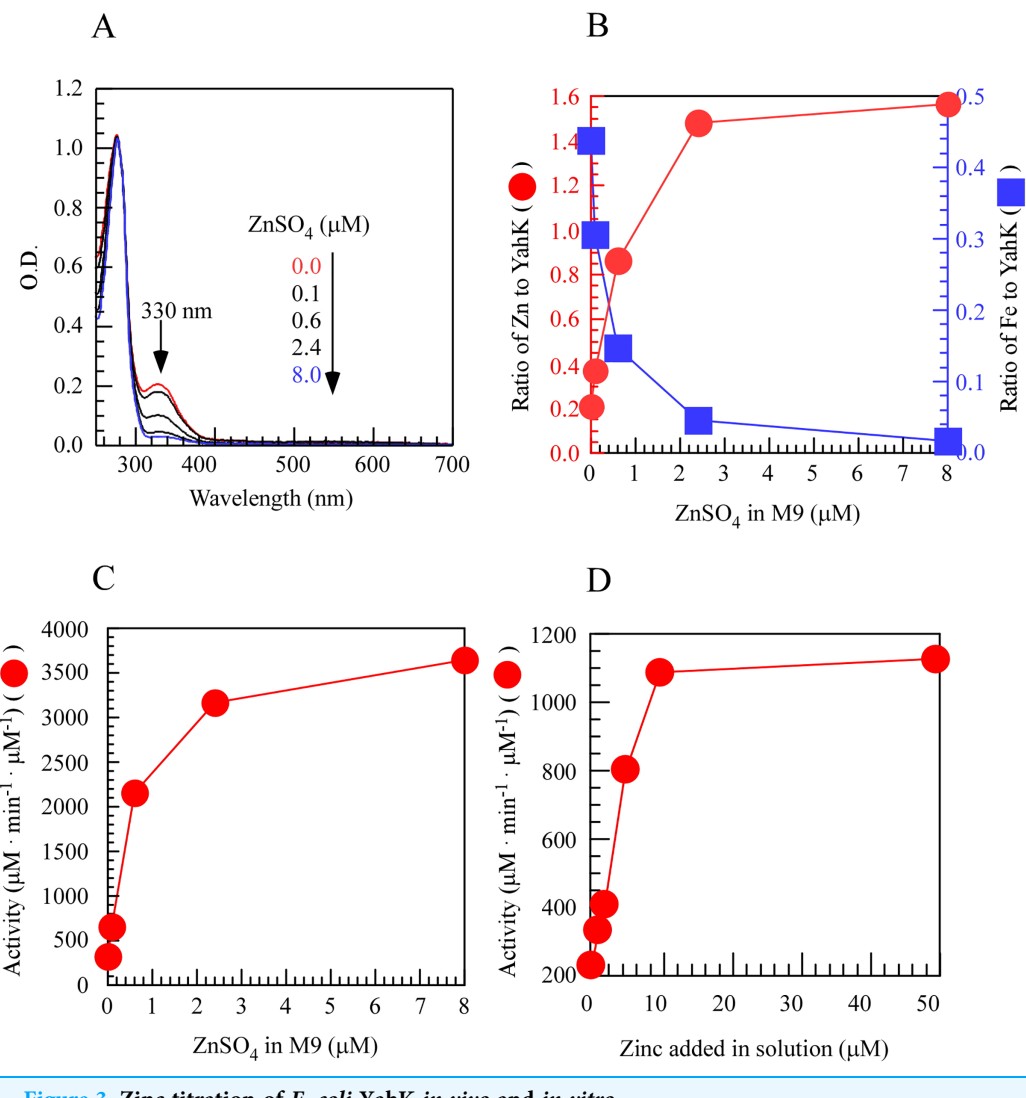

**Figure 3** Zinc titration of *E. coli* YahK *in vivo* and *in vitro*.

indicating substantial activation of YahK in the presence of adequate zinc (Figs. 3A–3C). *In vitro* experiments also support this idea (Fig. 3D). Additionally, in *E. coli* cells grown in the M9 minimal medium with a fixed zinc concentration (1 μM), we attempted to use excess iron to compete with zinc for binding to YahK. However, even when the iron concentration gradient in the medium was increased to 100 μM, we were unable to obtain completely metal-free Fe-YahK, and the results of the *in vitro* experiment were consistent with this (Fig. S1). Taken together, the results suggested that the metal-binding properties and catalytic activity of YahK are significantly influenced by the specific metal content in the culture medium.

## Localization of the iron-binding site of YahK

To elucidate the iron-binding sites of YahK, we constructed YahK mutants in which the active zinc-binding site Cys-40 and the structural zinc-binding site Cys-99 were replaced

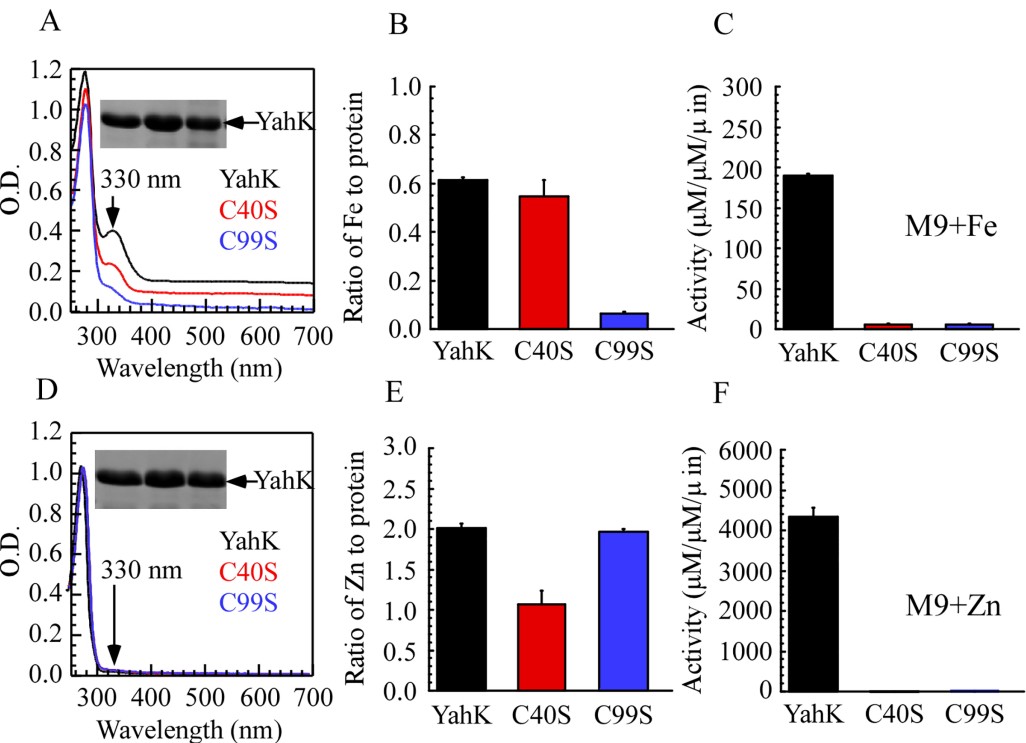

**Figure 4 Iron/zinc binding activity of *E. coli* YahK and YahK mutants.** (A) UV-visible absorption spectra of YahK (black lines) and YahK mutants (red and blue lines) purified from *E. coli* cells grown in M9 minimal medium supplemented with 50 μM ferric citrate. The protein concentration of the proteins was about 37.5 μM. The insert is a photograph of the SDS-PAGE gel of purified proteins. (B) The iron content of *E. coli* YahK and YahK mutants purified from *E. coli* cells grown in M9 minimal medium supplemented with 50 μM ferric citrate. (C) The activity of *E. coli* YahK and YahK mutants purified from *E. coli* cells grown in M9 minimal medium supplemented with 50 μM ferric citrate. (D) UV-visible absorption spectra of YahK (black lines) and YahK mutants (red and blue lines) purified from *E. coli* cells grown in M9 minimal medium supplemented with 50 μM zinc sulfate. The protein concentration of the proteins was about 37.5 μM. The insert is a photograph of the SDS-PAGE gel of purified proteins. (E) The zinc content of *E. coli* YahK and YahK mutants purified from *E. coli* cells grown in M9 minimal medium supplemented with 50 μM zinc sulfate. (F) The activity of *E. coli* YahK and YahK mutants purified from *E. coli*. The data are presented as the mean ± S.D. $p < 0.05$.

with serine (S). Wild-type YahK and its mutants were then expressed in *E. coli* cells grown in M9 minimal medium supplemented with $ZnSO_4$ or ferric citrate (50 μM), followed by purification of each protein from the *E. coli* cells. As illustrated in Figs. 4A–4C, the YahK-C40S mutant expressed in M9 minimal medium supplemented with 50 μM ferric citrate exhibited a lower absorption peak at 330 nm compared to the wild-type YahK, yet still maintained a distinct peak. The iron content in YahK-C40S was comparable to that of wild-type YahK. In contrast, the YahK-C99S mutant showed minimal absorption at the 330 nm peak, demonstrating significantly reduced iron affinity. Both C40S and C99S mutations markedly decreased the enzymatic activity of YahK.

Additionally, when 50 μM $ZnSO_4$ was added to the M9 minimal medium, zinc content in the YahK-C40S mutant was observed to be only half of that in wild-type YahK, whereas the zinc content in YahK-C99S nearly matched the levels found in wild-type YahK. The
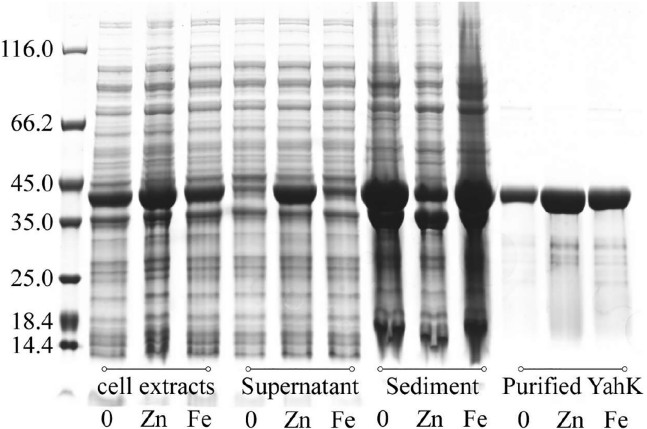

**Figure 5 Iron atom improves the solubility and stability of the target protein YahK to some extent.**

alcohol dehydrogenase activity of both mutants, YahK-C40S and YahK-C99S, was almost completely abrogated (Figs. 4D–4F).

The ablation of the active zinc-binding site Cys-40 predominantly affects zinc binding and the corresponding biological activity, while exerting minimal impact on iron binding. Conversely, mutation of the structural zinc-binding site Cys-99 substantially impairs YahK's iron-binding capacity and enzymatic activity. These findings illustrate that the Cys-99 residue is critical for iron binding and is involved in regulating both iron and zinc binding. Collectively, these experiments underscore the essential roles of Cys-40 and Cys-99 residues in the functional mediation of YahK.

### Iron maintains the stability of YahK

We expressed YahK in *E. coli* cells cultured in M9 minimal medium supplemented with either 50 μM ferric citrate or 50 μM zinc sulfate. Subsequently, the bacterial cells were lysed (with consistent turbidity) using sonication followed by centrifugation to separate the soluble proteins (supernatant) from the insoluble proteins (sediment, potentially containing inclusion bodies). Whole-cell protein electrophoresis analysis was then performed, and the gel was stained with Coomassie Brilliant Blue. As depicted in Fig. 5, compared to YahK induced for expression in zinc sulfate-supplemented medium, a significantly higher amount of YahK was observed in the sediment when cultured in both iron citrate-supplemented medium and metal-free M9 minimal medium, suggesting a propensity of YahK to form inclusion bodies during overexpression under these conditions.

### DISCUSSION

Trace elements are essential nutrients for the human body, and their presence is closely related to human health and the development of diseases (*Choi et al., 2018*; *Yadrick, Kenney & Winterfeldt, 1989*). Iron and zinc, as essential trace elements, play crucial roles in human health and various enzymatic processes. Zinc, in particular, serves as a component or activator for numerous enzymes, including alcohol dehydrogenase,

DNA topoisomerase, and RNA polymeras (*Abelein, Gräslund & Danielsson, 2015*; *Duncan & Hurley, 1978*; *Lu et al., 2011*; *Omichinski et al., 1993*). ADH is a class of oxidoreductases that employ $NAD^+$ or NADP as coenzymes to catalyze the reversible conversion of alcohols to aldehydes or ketones (*Zakhari, 2006*). It is widely distributed in various organisms, with microbial alcohol dehydrogenases gaining attention for their applications in synthesizing or modifying high-value alcohols, especially chiral alcohols, for use in fermentation engineering (*Semenova et al., 2020*).

The *E. coli* alcohol dehydrogenase YahK studied in this project belongs to the medium-chain dehydrogenase/zinc-dependent alcohol dehydrogenase family. However, the mechanism regulating its activity remains unclear, and the influence of metal ions on its activity needs further investigation. Previously, we unexpectedly discovered that *E. coli* topoisomerase I (TopA) is a novel iron/zinc binding protein whose activity is regulated by iron and zinc (*Lu et al., 2011*), prompting us to explore whether *E. coli* alcohol dehydrogenase YahK also binds iron and zinc. Here, we reported that YahK, purified from *E. coli* cells grown in LB medium, contains both iron and zinc, which differs from the reported property of recombinant *E. coli* YahK, which binds two $Zn^{2+}$ ions per subunit. Furthermore, the recombinant YahK expressed in M9 minimal medium exhibits markedly low levels of iron and zinc. However, upon supplementation with exogenous ferric citrate/zinc sulfate, the YahK demonstrates distinct iron or zinc binding capabilities. Unlike the zinc-binding *E. coli* alcohol dehydrogenase YahK, the iron-binding form of YahK almost lost its ability to catalyze ethanol dehydrogenation, indicating that iron-binding affects the occurrence of zinc-dependent enzymatic reactions.

The similar chemical properties of iron and zinc result in their ability to employ similar binding ligands in metal coordination chemistry. This leads to a certain level of competition bbetween these metals, as noted in previous studies (*Dauter et al., 1996*). Such competition can influence the catalytic function of important enzymes. For instance, the iron-bound form of *E. coli* topoisomerase I (TopA) loses its DNA unwinding activity, and the zinc finger of the GATA transcription factor family member, GATA-1 protein, exhibits higher activity in its iron-bound state than in its zinc-bound form (*Omichinski et al., 1993*).

Our research uncovered that as the concentration of exogenous zinc ions in M9 minimal medium increased, the characteristic absorption peak produced by iron-binding proteins at 332 nm decreased correspondingly. Metal content assays validated a decrease in iron binding, alongside a gradual increase in zinc binding, resulting in enhanced activity. The above phenomena indicated that YahK protein binds metal iron and metal zinc and there is a competitive relationship between iron and zinc. These findings prompted us to explore the mechanism of iron-mediated repression of YahK activity. Through site-directed mutagenesis experiments, we found that iron predominantly competes with zinc for binding to the cysteine residue at position 99 in the YahK amino acid sequence. In this context, when a mutation occurs at the cysteine residue at position 99, the protein's ability to bind iron is significantly diminished. This indicates that the structural zinc-binding site of YahK is involved in the binding of iron to YahK in *E. coli* cells. This

study confirms that YahK protein has a strong iron-binding ability, and iron could inhibit ADH activity. However, further research is needed to uncover the specific mechanism by which iron regulates the enzymatic activity of YahK.

The present study also revealed variations in the concentration of purified YahK protein in M9 minimal medium, with or without the presence of metal ions. Through initial SDS-PAGE analysis, we found that the YahK induced in M9 minimal medium without or only containing ammonium ferric citrate mainly exists in the form of inclusion bodies. Since the YahK protein of *E. coli* can bind zinc and iron, and iron mainly binds to its structural zinc site, it is inferred that iron plays an important role in substituting zinc and assuming a crucial conformation in YahK protein. Although this method cannot provide quantitative data, it is sufficient to provide initial visual evidence of the protein aggregation tendency. Furthermore, in our investigations, we discovered that *in vitro* co-incubation of Fe-YahK and a gradient of zinc enabled the protein to bind to zinc ions and restore its alcohol dehydrogenase activity. However, it is noteworthy that *in vitro* incubation of iron ions with Zn-YahK yielded unsatisfactory result, necessitating the combination *in vivo* by introducing iron ions into the M9 minimal medium. This suggests that the interaction between *E. coli* YahK and iron ions may require additional molecular chaperones or a specific *in vivo* environment. Consequently, it becomes imperative for us to investigate comprehensive studies pertaining to the dynamic binding process of YahK and iron ions *in vivo*. This will contribute to a more thorough understanding of the physiological and pathological significance of YahK's iron-binding capabilities.

Furthermore, human alcohol dehydrogenase ADH1B belongs to the homologous protein of *E. coli* alcohol dehydrogenase I (zinc-dependent medium and long-chain ADHs). The enzyme activity is more significant in the Han Chinese population (*Peng et al., 2010*). Studies showed that the single nucleotide polymorphism of its gene was associated with various diseases, such as digestive system diseases, neurodegeneration, and metabolic diseases (*Almeida et al., 2020*; *Im et al., 2022*; *Zimta et al., 2019*). In the field of protein sequence alignment analysis, human ADH1B exhibits a 26% homology and 40% similarity in its sequence with *E. coli* YahK. Further examination reveals that the sequence of the human ADH1B contains an identical zinc-binding motif to that of the *E. coli* YahK, specifically $GHEX_2GX_5$ (G/A)$X_2$ (I/V/A/C/S). This implies that despite significant evolutionary divergence, there is a certain degree of conservation within their functional structural domains (Fig. S2). Hence, it is necessary for us to explore the iron and zinc binding ability of human ADH1B and the mechanism of alcohol dehydrogenase activity in our future studies. These studies will contribute to a deeper understanding of human iron and zinc metabolism and its correlation with diseases, offering valuable insights for medical research.

## CONLUSION

Our research investigates the metallophilic properties of *E. coli* alcohol dehydrogenase YahK and elucidates their mechanistic implications on enzymatic activity. The results demonstrate that YahK not only binds zinc but also exhibits a robust capacity for iron binding, which significantly affects its activity. Interestingly, YahK maintains and

sometimes surpasses its iron-binding affinity even in the iron-deficient M9 medium compared to the LB medium. Further quantification of metal content reveals a competitive relationship between iron and zinc binding on YahK: increased iron binding corresponds to decreased zinc binding, and *vice versa*. This competitive dynamic is supported by a series of *in vivo* and *in vitro* experiments. Moreover, we discovered that iron exerts an inhibitory effect on the enzyme activity of YahK; higher levels of iron binding result in lower alcohol dehydrogenase (ADH) activity. To unravel this inhibition mechanism, site-directed mutagenesis was performed, revealing that iron primarily competes with zinc for the cysteine residue at the 99th position of the amino acid sequence. Mutation at this site significantly reduces iron binding; however, as this site also stabilizes protein structure through zinc coordination, altering it potentially renders the protein inactive due to structural instability.

The dynamic interplay between iron and zinc binding within YahK sheds light on its intricate regulatory mechanisms as an alcohol dehydrogenase and underscores its potential physiological role in *E.coli* cellular metabolic regulation. This discovery opens new avenues for exploring the specific impacts of metal ion binding on both YahK and *E.coli* cellular metabolism. In summary, it is evident that YahK has a pronounced affinity for binding with iron, which imposes an inhibitory effect on ADH activity. However, further investigation is required to fully understand the exact mechanisms through which iron modulates Yahk's activity.

## ACKNOWLEDGEMENTS

We thank Editage for its linguistic assistance during the preparation of this manuscript.

### Funding

This work was financially supported by the Wenzhou Municipal Science and Technology Bureau of China (Y20220743) and Key Laboratory of Clinical Laboratory Diagnosis and Translational Research of Zhejiang Province (2022E10022). The funders had no role in study design, data collection and analysis, decision to publish, or preparation of the manuscript.

### Grant Disclosures

The following grant information was disclosed by the authors:
Wenzhou Municipal Science and Technology Bureau of China: Y20220743.
Clinical Laboratory Diagnosis and Translational Research of Zhejiang Province: 2022E10022.

### Competing Interests

The authors declare that they have no competing interests.

## Author Contributions

- Feng Liang conceived and designed the experiments, performed the experiments, analyzed the data, prepared figures and/or tables, authored or reviewed drafts of the article, and approved the final draft.
- Shujuan Sun conceived and designed the experiments, performed the experiments, analyzed the data, prepared figures and/or tables, authored or reviewed drafts of the article, and approved the final draft.
- YongGuang Zhou performed the experiments, analyzed the data, prepared figures and/or tables, and approved the final draft.
- Tiantian Peng analyzed the data, prepared figures and/or tables, and approved the final draft.
- Xianxian Xu analyzed the data, prepared figures and/or tables, and approved the final draft.
- Beibei Li analyzed the data, prepared figures and/or tables, and approved the final draft.
- Guoqiang Tan conceived and designed the experiments, authored or reviewed drafts of the article, and approved the final draft.

## Data Availability

The raw data is available in the Supplemental File.

## Supplemental Information

Supplemental information for this article can be found online at http://dx.doi.org/10.7717/peerj.18040#supplemental-information.

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
