# Peer review of "Escherichia coli alcohol dehydrogenase YahK is a protein that binds both iron and zinc"

_PeerJ, doi:10.7717/peerj.18040_

## Round 0.1 · original submission · Major Revisions

Three experts assessed your manuscript and raised several concerns that need to be addressed. Among the relevant ones is the lack of sufficient description of methodologies, hypotheses, and rationales, and insufficient raw data to support results.

Reviewer 1 ·

Basic reporting

Dear Authors,
This study has put good efforts for studying the activity regulation of E. coli alcohol dehydrogenase in relation to to iron and zinc concentrations.
I suggest some parts of the article need to be more clear.

1- Abstract section could be improved by adding the important findings of the work.
Plus, according to the journal instruction: “Headings in structured abstracts should be added in bold and followed by a period”. Here, no headings are added. Add subheadings (Background, Objective, Methods, Results, and Conclusion). In addition the main hypothesis to be tested is not well-constructed.

2- Introduction section: Some sentences need reference to be added. It lacks well-constructed aim of the work.
3- M & M section: Lack statistical analyses description.
4- Results section: Some paragraphs do not belong to results and should be transferred to M & M section or to Discussion section. Many paragraphs need to be rephrased and clarified.
5- Discussion section: Some paragraphs do not belong to results and should be transferred to M & M section.
6- Conclusion section: Lack conclusion section.
7- References section: Should be revised according to PeerJ format.
8- English language and style: English language need to be improved throughout the whole MS. I suggest it to be revised by a fluent English speaker.

The overall workload is good but the specific hypothesis to be tested is not well constructed, and with some improvement it will be better.

In my opinion, all figures are presented well and some of which need little re-work.

Experimental design

The authors analyzed the activity regulation of E. coli alcohol dehydrogenase in relation to to iron and zinc concentrations.
Although the authors presented important findings, the manuscript need a further review.

1- The purpose of the study is not well defined neither in abstract nor in introduction.
2- In methods section, it lacks description of statistical analyses done. Some other methodologies need more details, clarification, and references. Line by line review is provided in the annotated pdf file, you will find some suggestions.

Validity of the findings

1- In Results section, although the authors did statistical analyses “as I understood”, there is no reflection of these analyses in writing the results. PLZ present the results in the light of your statistical analyses.
2- Also, the results should be described as they are presented in Figs and Tables. You should describe your finding without speculations in Results section.
3- In addition some sentences in the Results belong to discussion section (Interpretations and speculations). Other paragraphs lack clear presentation and use very long sentences and hard to read (kindly rephrase). Other paragraphs belong to M & M section (those interpreting methods or why we did such analyses).
4- Finally, figures are well-constructed but lack detailed illustrated legends. (See line by line review in the pdf file).
5- In Discussion section, the same comment on Results section. Some paragraphs lack clear presentation and use very long sentences and hard to read (kindly rephrase). Other paragraphs belong to other sections (Kindly put in their proper section). I included some specific comments in the pdf (See line by line review in the pdf file).
6- The article lacks Conclusion section.

Additional comments

1- PLZ consider that the abstract in PeerJ format including subheadings (Background, Objective, Methods, Results, and Conclusion). Headings in structured abstracts should be added in bold and followed by a period.
Also you have to construct the main objective of your work in both Abstract and Introduction sections. Additionally, make the subsequent required corrections.
2- In Introduction, Please add the required references and make the subsequent corrections.
3- Add statistical analysis topic to the M & M section and describe the statistical methods and data manipulation.
4- In Results make the required corrections to your Figures and Tables.
Describe your results in the light of your well-presented Figures, and statistical analyses you did.
5- Legend of the Figure should be well-illustrated. Showing what A, B, C, D, and E panels are???? and what are they illustrate? This is true for all your Figures and Tables.
Specific comments on Figures:
Figures 1 & 3: It will be better if you write the “Title of the axes in black instead of colored”.
Figure 2: It will be better if you delete 1, 2, 3, from the figure (See pdf).
Figure 4: It will be better if you make it colored instead of white-black figure.
Table 1: It will be better if you include all primers in this table, and explain which primer is used in which experiment and for which purpose? In addition the site-directed mutation primers should be noted in M & M section, and all subsequent corrections should be made.
6- In Results section, just describe your findings without interpretation or speculations.
7- In discussion section, interpret your results, compare with the previous results in the subject (agree or disagree), give reasonable interpretation to the disagreeable research, and finally extract your definite conclusion.
8- References should be revised in accordance with the PeerJ format.

Annotated reviews are not available for download in order to protect the identity of reviewers who chose to remain anonymous.

Reviewer 2 ·

Basic reporting

Unclear and ambiguous language used throughout.
Line 26 – 44: The abstract does not define research questions; results are expressed with speculative language such as ‘seems to enhance’ (line 36), ‘could compete’ (line 38), ‘may be involved’ (line 40). The last sentence of the abstract (line 42 – 44), completely devalues the findings described in the line 32 - 42. The last line should emphasize the most important discovery of the manuscript.
Speculative languages are used to describe results throughout the article. Examples, ‘to some extent’ (line 183, 260, 280, figure 5), ‘to a certain extent (line 269, 277). I would suggest providing quantitative data to support the statements.
Experimental evidence should be described as facts, and there is a room for speculation in the conclusions section with note that future investigations are required to ascertain that speculation.
Intro & background do not show context—Introduction lacks sufficient description of E. coli aldehyde/alcohol dehydrogenase. It is not clear to me what is the motivation that inspired the authors to do biochemical characterization of YahK.
Important references are missing –
1. Nguyen et. al. 2020 - Structural Basis for Broad Substrate Selectivity of Alcohol Dehydrogenase YjgB from Escherichia coli.
2. Sevcenco et. al. 2011, Exploring the microbial metalloproteome using MIRAGE
Line 64 -67: The references provided are not sufficient to prove the zinc binding to those amino acids. Please add a protein sequence alignment showing those amino acids and provide appropriate reference or prediction modeling.
Figures are relevant, not high quality, not well labeled & described.
Raw data supplied but lack enough metadata to understand which table corresponds to which figure.
Line 40 – It is not clear to me what the multifaceted nature of the enzyme YahK.

Experimental design

Research questions and hypotheses are not well defined.
It does not state how the research fills an identified knowledge gap.
Method descriptions are insufficient — please follow the material method section of Pick et. al. 2013 and Nguyen et. al 2020.
Line 89 – 93 – The primers lack restriction enzymes HindIII and NcoI. The cloning direction in pET28b+ is not clear – is it a C-terminal or N-terminal His tag?
Individual Fe or Zn metal binding affinities and stoichiometry to YahK should be directly determined and compared. The calculation of Kd will be necessary to confirm the binding specificity of the enzyme.
Figure 3 addresses the effect of cofactor deficiency on YahK activity. YahK is a bifunctional aldehyde and alcohol dehydrogenase. It might be worthwhile to investigate the effect of excess substrate (ethanol or aldehyde) on YahK activity with respect to iron binding effect. The authors also mentioned in the discussion that zinc binding enhances the ADH activity. Could iron binding enhance the aldehyde reductase activity of YahK?
For every substrate-enzyme kinetic study please calculate Km and Kcat value

Validity of the findings

This manuscript is an experimental replication of study done by authors on TopA “Escherichia coli topoisomerase I is an iron and zinc binding protein”.
The rationale & benefit to literature is not clearly stated in the context of YahK. For example, not enough background was given for alcohol/aldehyde dehydrogenase. There are several alcohol/aldehyde dehydrogenase studies in E. coli.
All underlying data are not provided; they are not robust, statistically sound, & controlled. Example fig 3D is not given in the manuscript.
Conclusions are not well stated & not limited to supporting results.
Line 209 and 362 – Data not shown- I would recommend providing the data or explaining why the data are not available.
Line 206 -212 – The conclusions are based on data that is not given in this manuscript. Fig 2 shows iron and zinc binding in YahK even without the external supplementation of Fe and Zinc.
Line 223 – 226 – There is no figure 3D in the figure section.
The data suggest Zn metal affects solubility, because most of the protein is detected in sediment fraction (fig 5). Authors mentioned stability – this requires further explanation.
Line 267 – 269 – This conclusion needs evidence from expressing both the mutated (C40A, C99A) and wild type protein in presence and absence of Zn and Fe.
Line 311 – “Protease” – It is a dehydrogenase.
Line 321 – 326 – There are repeated sentences here.
Line 330 – 336 – This conclusion is not supported by evidence rather speculation.
Line 355 – 372: These are purely speculations and subjects of future study. Please provide sequence alignment of Yahk with these proteins showing conserved amino acid residues mutated in the research.

Reviewer 3 ·

Basic reporting

I am very much delighted to review the manuscript (#93693) and found that the authors have put their valuable efforts and knowledge to do an innovative research and have prepared the manuscript nicely.

The manuscript has been written following clear English language and grammar. The Introduction chapter contains relevant background literature and well-referenced information. All the figures, table and data have been provided which are statistically measured and presented.

Experimental design

Experimental design and methods are well described. The analytical parameters are adequate for the interpretation and discussion of the results. Research questions are well-defined and explored. This is an original research paper so far. Major results showed how the in vivo regulation of Escherichia coli YahK involve the binding of both iron and zinc and its contribution in cellular metabolism.

Validity of the findings

The conclusion summarizes findings, and discusses the implications and future research directions. However, I have suggested minor revisions marked in the manuscript.

Annotated reviews are not available for download in order to protect the identity of reviewers who chose to remain anonymous.

---

## Round 0.2 · Minor Revisions

The manuscript was improved, but there are issues with the statistical analyses. Please address the comments raised by the Reviewer.

Reviewer 1 ·

Basic reporting

Generally, I see that the MS has been significantly improved. I appreciated the efforts of the authors to reach out to the present form of the MS.
For me the present form is generally satisfactory and addressed most of my criticisms.

Experimental design

Satisfactory improved

Validity of the findings

The only criticisms that I have are that the following are NOT addressed:
1- In Results section, although the authors did statistical analyses “as I understood”, there is no reflection of these analyses in writing the results. PLZ present the results in the light of your statistical analyses.

NOT Addressed. I DON'T SEE significance level of increase or decrease (P=???) in the results section, .......etc. Kindly note that if you just mentioned the numerical value of a figure or table, this is just REDUNDANCY in the result section.
You have to mention what is beyond the numerical values in tables and figures.
Is a value increased, decreased or not, Is a relation between the factors in the study, direct or inverse relation ......etc. Also is this change (increase or decrease) statistically significant (P=???) or insignificant (P=???).
You have to mention the physical meaning of the numerical results, avoiding repeatedness of the results in Tables and Figures.
ALSO I commented on:
Figures 1 & 3: It will be better if you write the “Title of the axes in black instead of colored”.
I asked to convert only the “Title of the axes in black instead of colored”. BUT you turned all the figs to W & B color. Kindly,
1- Return to the colored Figure 1 and Just WRITE The words "YahK, AdhP" in Panels B, C, and D of Figure 1 IN BLACK.
2- Return to the colored Figure 3 and Just WRITE The words "Ratio of Zn to YahK, Ratio of Fe to YahK" in Panels B of Figure 3 IN BLACK.
3- Return to the colored Figure 2 and Just DELETE the numbers as in my comment.
4- Figure 4: It will be better if you make it colored instead of white-black figure.

Additional comments

For me the present form is generally satisfactory and addressed most of my criticisms.
It just need very minor revision.
PLZ address my requested improvements in the above section.
Thank you

---

## Round 0.3 · accepted · Accept

The authors addressed all the pending observations and the new manuscript version is now suitable for publication.